# On the Quality Control of HY-2 Scatterometer High Winds

Shuyan Lang [1,2], Wenming Lin [2,3,]*, Yi Zhang [1,2] and Yongjun Jia [1,2]

[1] National Satellite Ocean Application Service, Beijing 100081, China
[2] Key Laboratory of Space Ocean Remote Sensing and Application, Ministry of Natural Resources, Beijing 100081, China
[3] School of Marine Sciences, Nanjing University of Information Science and Technology, Nanjing 210044, China
[*] Correspondence: wenminglin@nuist.edu.cn

**Abstract:** The operational wind processor for the Ku-band scatterometers onboard HY-2 satellite series uses a quality control (QC) scheme based on the maximum likelihood estimator (MLE). Since it is difficult to discriminate rain contamination from "true" high winds, the MLE-based wind QC is set in a conservative way, which rejects up to ~35% of high winds ($w \geq 20$ m/s) in HY-2 scatterometers (HSCATs). In this paper, the sensitivity of MLE and its spatially averaged value (i.e., $\mathrm{MLE_m}$) to wind quality and rain is reconsidered by analyzing the collocated HSCAT observations and buoy data, as well as rain data from the global precipitation measurement satellite's microwave imagers. It shows that $\mathrm{MLE_m}$ is more effective than MLE in terms of flagging rain data. More interestingly, the HSCAT high winds are much less strongly affected by rain, compared to the prior Ku-band pencil-beam scatterometers (e.g., RapidScat). Consequently, a $\mathrm{MLE_m}$-based approach is proposed to improve the HSCAT wind QC, particularly for high winds. The new QC method results in ~8% rejections at 20 m/s and above. Compared to the collocated buoy winds, the HSCAT high winds preserved by the new QC (but rejected by the operational QC) are of fairly good quality.

**Keywords:** scatterometer; high winds; quality control; maximum likelihood estimator (MLE); HY-2 satellite

## 1. Introduction

The Ku-band pencil-beam scatterometers onboard the Haiyang-2 (HY-2) satellite series are designed to observe near-surface vector winds over the globe's oceans [1]. At present, the HY-2 constellation consists of three satellites, with HY-2B operating in a sun-synchronous orbit, and HY-2C and -2D operating in drifting orbits of 66° inclination. As such, the HY-2 satellite scatterometers (HSCATs) provide global sea-surface wind observations with unprecedented short revisit intervals. Moreover, recent advances in the HSCAT data processing show that the radar backscatter measurements of the three HSCATs are well inter-calibrated, leading to consistent and high-quality retrieved winds [2–4].

However, as was the case with previous Ku-band pencil-beam scatterometers, e.g., QuikSCAT [5], RapidScat [6], the Oceansat-2 scatterometer (OSCAT) [7], rainfall is the primary factor in degrading the HSCAT wind quality. Consequently, a quality control (QC) scheme is needed to discriminate good-quality winds from rain-contaminated ones, in order to achieve a successful use of HSCAT winds in a variety of applications, such as nowcasting, disaster monitoring, and numerical weather prediction (NWP) data assimilation, among others. In practice, the HSCAT wind QC is adapted from the Pencil-Beam Wind Processor (PenWP) provided by the European Organization for the Exploitation of Meteorological Satellites (EUMETSAT) Ocean and Sea Ice (OSI) Satellite Application Facility [8]. That is, the inversion residual of the maximum likelihood estimator (MLE), which is commonly used in scatterometer wind retrieval, is adopted to discern between good- and poor-quality winds. The MLE-based QC has been proven to be good but not optimal for flagging rain data in Ku-band scatterometers [9,10]. Hence, the MLE threshold is operationally set in a conservative

way, leading to ~1% rejections at very low wind speeds, and up to ~35% rejections at 20 m/s and above [11]. Overall, 5–6% of HSCAT winds are currently rejected, which is much higher than the ~0.5% rejection rate associated with C-band scatterometers [12].

In recent years, several QC indicators other than MLE have been proposed to improve rain discrimination in Ku-band scatterometers [6,9], with the objective of preserving as many valuable wind observations as possible, notably for high wind data. For instance, Lin and Portabella [3] showed that both the spatially averaged MLE value over multiple wind vector cells (i.e., 3 × 3 box) and the singularity exponents (SE) derived from the so-called singularity analysis technique [13] are more sensitive to the quality of the retrieved winds, compared to the conventional MLE indicator. Xu and Stoffelen [9] used the speed difference between the two-dimensional variational analysis (2DVAR) wind field and the scatterometer-selected ambiguous solution (namely $J_{oss}$) to reduce the false alarm rate associated with the conventional MLE QC. Moreover, different QC indicators may be complementary in detecting rain or categorizing wind quality for Ku-band scatterometers; as such, they are usually combined to further improve rain detection using a multidimensional histogram technique [10,14].

So far, the above progress on the wind QC of Ku-band scatterometers has rarely been adapted for the HY-2 satellite scatterometers. Considering that the HSCATs observe the Earth's surface in slightly lower incidence angles (i.e., 41.5° for the horizontally polarized (HH) beam, and 48.5° for the vertically-polarized (VV) beam) than the preceding pencil-beam scatterometers, e.g., QuikSCAT (46° for the HH beam, and 54° for the VV beam), RapidScat (49° for the HH beam, and 56° for the VV beam), and OSCAT (49° for HH beam, and 58° for VV beam), etc., it is necessary to carry out a comprehensive study on the HSCAT wind QC. In this paper, the performance of the operational MLE-based QC is evaluated for the HSCATs onboard HY-2 satellite constellations. Then, an improved QC method is proposed to mitigate the over-rejection rates at high wind conditions. Section 2 describes the datasets and the methods used in this study. Section 3 provides an overview of the HSCAT MLE-based QC. The effectiveness of the proposed QC method is examined in Section 4 by analyzing the collocated HSCAT and rain data (or buoy wind data). Finally, the main conclusions are summarized in Section 5.

## 2. Data and Methods

### 2.1. Data

The HSCATs' 25-km level 2B (L2B) data from the National Satellite Ocean Application Service (NSOAS) were used to assess the performance of different QC methods. This dataset consists of three years of HSCAT-B data (2019–2021), 14 months of HSCAT-C data (October 2020–December 2021), and seven months of HSCAT-D data (June–December 2021). Sea surface winds derived from different HSCATs were assessed in [2–4], which showed that the HSCAT winds are of good quality, and are consistent among different satellite platforms. Note that the HSCATs L2B files already include the European Centre for Medium-Range Weather Forecasts (ECMWF) wind data, which were acquired by interpolating three ECMWF three-hourly forecast winds, both spatially and temporally, to the HSCATs observation location and time, and were used as the background wind field in the ambiguity removal processing. For the purposes of this study, the ECMWF winds were used in the wind quality assessment for the HSCATs.

To evaluate the sensitivity of QC indicators to rain, the above HSCAT observations were collocated with the global precipitation measurement (GPM) satellite's microwave imager (GMI) rain data. With collocation criteria of less than 30 min and 0.125° spatial distance between the HSCAT and the GMI acquisitions, more than one million GMI rain measurements were collected every month for each HSCAT instrument. As such, the amount of HSCAT-GMI collocations is sufficient to perform a comprehensive sensitivity analysis in the next section. Figure 1 shows the spatial distribution of the HSCAT-B and GMI collocations, and the GMI mean rain rate in 2021. Note that the collocations are more likely to be acquired around the latitudes of ±60°, while the rain contamination mostly

appears at mid and low latitudes. Therefore, the percentage of QC rejected data, or the rejection ratio, of the HSCAT-GMI collocations may be smaller than the overall HSCAT dataset.

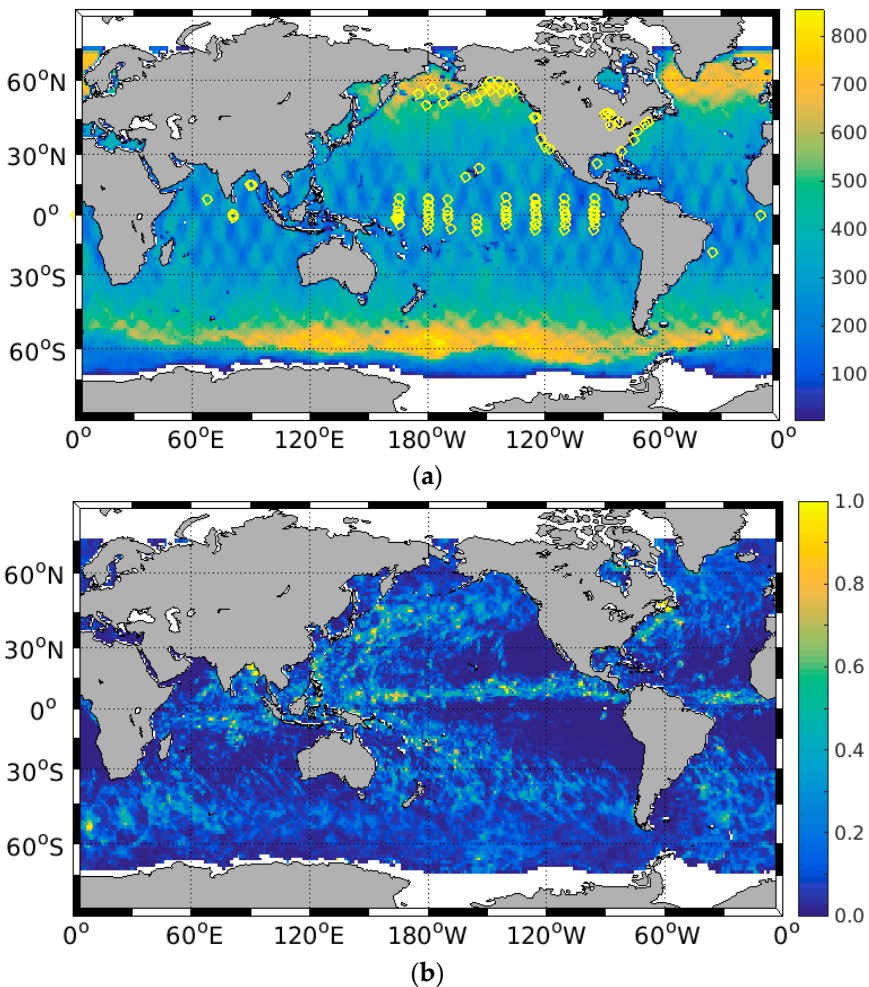

**Figure 1.** Spatial distribution of (**a**) the HSCAT-B—GMI collocations and (**b**) the GMI mean rain rate on a global map. Here, the collocations for 2021 are used. Color bars indicate the number of collocations (**a**) and the mean rain rate (mm/h). The yellow circles indicate the locations of the moored buoys used in this study.

Additionally, the HSCAT observations were also collocated with the moored buoy winds in order to verify the wind quality associated with different rain flags. The buoy wind data are of 10 min temporal resolution, which can be further used to compute 25-km equivalent buoy winds (i.e., mean buoy winds), following Taylor's hypothesis [15,16]. As shown in [16], the mean buoy wind is more representative of the scatterometer area-mean wind (e.g., 25 km spatial resolution) than a 10 min buoy measurement; therefore, the mean buoy winds were used as a reference in the following analysis. The buoy acquisition time and position are with 5 min and a 25 km distance of the HSCAT observation. The total amount of collocations with buoy wind data is about 53.6 k for HSCAT-B, 21.3 k for HSCAT-C, and 14.5 k for HSCAT-D. For the sake of illustration, the buoy locations are plotted in yellow circles in Figure 1a.

*2.2. Methods*

In PenWP, the MLE value used for QC purpose is derived by minimizing the following equation:

$$\text{MLE} = \frac{1}{N} \sum_{i=1}^{N} \frac{\left(\sigma_{\text{m}i}^0 - \sigma_{\text{s}i}^0\right)^2}{\left(K_{\text{p}i}\sigma_{\text{m}i}^0\right)^2} \qquad (1)$$

where $N = 4$ is the number of views in a certain wind vector cell (WVC), $K_{\text{p}i}$ is the normalized measurement error of the $i$th beam, $\sigma_{\text{m}i}^0$ is the normalized radar backscatter coefficient, and $\sigma_{\text{s}i}^0$ is the corresponding backscatter value simulated through the NSCAT-4 geophysical model function (GMF) [17], which represents the empirical relationship between the sea-surface wind vector and the radar backscatter coefficient for various observation geometries. As such, MLE represents the distance of the measured radar backscatter values to the GMF used in the wind retrieval. Large MLE value indicates that the backscatter measurements are too far away from the GMF, in which case the corresponding WVC is flagged as a poor-quality retrieval. More specifically, the MLE threshold used to set the flag, which is wind-speed-dependent, was originally determined from the analysis of the collocated QuikSCAT and ECMWF data, and then adapted to reject the same percentile of data for the HSCATs.

A spatially averaged MLE ($\text{MLE}_\text{m}$) was proposed in [6] to account for the inter-WVC wind variability on the one hand, and to reduce the noise associated with the MLE field on the other hand. $\text{MLE}_\text{m}$ has been successfully used to improve the RapidScat wind QC. Since the $\text{MLE}_\text{m}$ value can be simply derived from the MLE field corresponding to the selected wind solutions, it is proposed as a means of flagging poor-quality L2B winds for the HSCATs. According to [6], $\text{MLE}_\text{m}$ is defined as follow:

$$\text{MLE}_{\text{m},i_0 j_0} = \frac{\sum\limits_{i=-1}^{1} \sum\limits_{j=-1}^{1} w_{i,j} \text{MLE}_{i_0+i,j_0+j}}{\sum\limits_{i=-1}^{1} \sum\limits_{j=-1}^{1} w_{i,j}} \qquad (2)$$

That is, the $\text{MLE}_\text{m}$ value of WVC $(i_0, j_0)$ is calculated from a centered $3 \times 3$ box (or a $3 \times 2$ box for WVCs along the swath edges). The objective of this paper is not to optimize the weighting scheme, so the $w_{i,j}$ values are set in the same way as [6].

$$w_{ij} = \begin{cases} 4, |i| + |j| = 0 \\ 3, |i| + |j| = 1 \\ 2, |i| + |j| = 2 \end{cases} \qquad (3)$$

This paper focuses on the HSCAT wind QC using MLE and $\text{MLE}_\text{m}$. The aforementioned indicators, namely SE and $\text{J}_{\text{oss}}$, cannot be estimated straightforwardly from the HSCAT L2B data, so they are studied below.

The sensitivity of MLE or $\text{MLE}_\text{m}$ to rain is assessed at different wind speed conditions. First, the HSCAT-GMI collocations are separated into 21 categories according to the retrieved wind speed ($w$) in bins of 1 m/s for $w \le 20$ m/s, apart from the last bin, which has $w > 20$ m/s. Second, the data in each category are ordered by the related HSCAT MLE/$\text{MLE}_\text{m}$ values in bins of 1%, i.e., the first bin contains the 1% highest MLEs/$\text{MLE}_\text{m}$s, the second bin the next 1% highest MLEs/$\text{MLE}_\text{m}$s, and so on. Afterwards, the percentile of GMI rain data above x mm/h (e.g., x = 0, 1, 2 . . . ) is computed for each bin. Eventually, the sensitivity of the QC indicators to rain can be easily evaluated based on the 2-D contour plot of the percentile of GMI rain data versus the retrieved wind speed and the indicator value.

It is well known that the wind retrieval skill of pencil-beam scatterometers is highly dependent on the number of independent views and their azimuth diversity, which both vary with the cross track distance (CTD) to the sub-satellite track. Consequently, the HSCAT

swath is assigned to three different regions (see Table 1) according to the CTD value, and then the analysis of the QC results is performed for different swath regions, in order to better understand the wind characteristics of HSCATs. Note that the outer swath is only observed by the HH beam.

**Table 1.** Swath categorization for the HSCATs.

| Name | Inner Swath | Sweet Swath | Outer Swath |
|---|---|---|---|
| CTD (km) | ≤200 km | 200–700 km | ≥700 km |
| WVC column number | 30–47 | 10–29 & 48–67 | 1–9 & 68–76 |

## 3. Overview of the MLE-Based QC and the Sensitivity of the QC Indicators to Rain

### 3.1. Statistical Analysis of the Operational MLE-Based QC

The overall performance of the HSCATs' operational QC is assessed by computing the difference between the HSCATs observations and the collocated ECMWF winds or buoy winds. The ECMWF winds, which are used for the ambiguity removal of HSCAT L2 processing, are already included in the L2B files. As such, three months (October–December 2021) of the HSCATs L2B data are adequate for the general quality assessment using ECMWF as reference. Nevertheless, all the HSCAT-buoy collocations in Section 2.1 are used for the assessment, using buoy winds as the reference.

Practically, the bias and the standard deviation (SD) of the scatterometer wind components (i.e., wind speed, direction, zonal ($u$) and meridional ($v$) components) with respect to (w.r.t.) the reference data are calculated and used to represent the wind quality. Table 2 shows the statistical scores of the three HSCATs w.r.t. ECMWF winds, whereas Table 3 shows the same scores w.r.t. the collocated buoy winds, in which the mean buoy wind vectors described in Section 2 are used in the comparison. The rejection ratio in the HSCAT-buoy collocations is higher than in the general assessment, because the moored buoys are generally located off the coasts of North America and in tropical oceans (see Figure 1a), where the sea-surface wind observations made by scatterometers are more likely to be contaminated by rain and high rates of wind variability [18].

**Table 2.** Standard deviations of the difference between HSCATs and ECMWF winds. The biases of HSCATs' wind components w.r.t. ECMWF are shown in the brackets.

| Statistical Scores | | Speed (m/s) | Direction (°) | $u$ (m/s) | $v$ (m/s) | Rejection Ratio (%) |
|---|---|---|---|---|---|---|
| QC accepted data | HSCAT-B | 1.13 (0.09) | 10.6 (0.5) | 1.23 (0.10) | 1.20 (0.02) | |
| | HSCAT-C | 1.07 (0.11) | 10.9 (0.7) | 1.23 (0.13) | 1.23 (0.01) | - |
| | HSCAT-D | 1.08 (0.16) | 10.4 (0.4) | 1.19 (0.16) | 1.21 (0.03) | |
| QC rejected data | HSCAT-B | 2.37 (1.30) | 17.7 (−0.1) | 2.56 (0.40) | 2.30 (0.07) | 5.74 |
| | HSCAT-C | 2.22 (1.16) | 17.3 (−0.1) | 2.40 (0.36) | 2.24 (0.05) | 5.85 |
| | HSCAT-D | 2.03 (0.93) | 14.4 (−0.1) | 2.10 (0.30) | 1.98 (0.05) | 9.80 |

**Table 3.** The same as Table 2, but using the mean wind vectors of the collocated buoy data as the reference.

| Statistical Scores | | Speed (m/s) | Direction (°) | $u$ (m/s) | $v$ (m/s) | Rejection Ratio (%) |
|---|---|---|---|---|---|---|
| QC accepted data | HSCAT-B | 0.92 (0.13) | 13.6 (1.4) | 1.47 (−0.07) | 1.39 (−0.10) | |
| | HSCAT-C | 1.10 (0.27) | 14.8 (−0.1) | 1.68 (−0.08) | 1.68 (−0.08) | - |
| | HSCAT-D | 1.02 (0.15) | 13.8 (−1.0) | 1.49 (−0.01) | 1.56 (−0.07) | |
| QC rejected data | HSCAT-B | 2.36 (0.89) | 31.9 (2.3) | 3.80 (0.20) | 3.29 (−0.32) | 7.49 |
| | HSCAT-C | 2.04 (0.63) | 25.9 (−1.9) | 3.47 (−0.04) | 3.11 (−0.56) | 7.95 |
| | HSCAT-D | 2.00 (0.79) | 24.7 (−0.8) | 2.93 (−0.27) | 3.44 (−0.34) | 11.00 |

Tables 2 and 3 prove that sea surface winds retrieved from different HSCATs are consistent and of fairly good quality (see the QC-accepted data). Moreover, the MLE-based QC discerns well between good-quality winds and poor-quality ones, though the rejection ratio of HSCAT-D appears abnormally higher than the other two HSCATs. The QC-rejected data are further evaluated in Figure 2, which shows the HSCAT-B rejected wind speed versus ECMWF (a) and buoy winds (b). It is clear that quite a few good-quality high wind observations ($w > 20$ m/s), which are of great significance to disaster monitoring, wind energy exploitation, data assimilation, etc., were rejected by the operational QC. Such a conclusion also holds for HSCAT-C and -D (not shown). This motivates the authors to improve the QC for HSCATs for high winds.

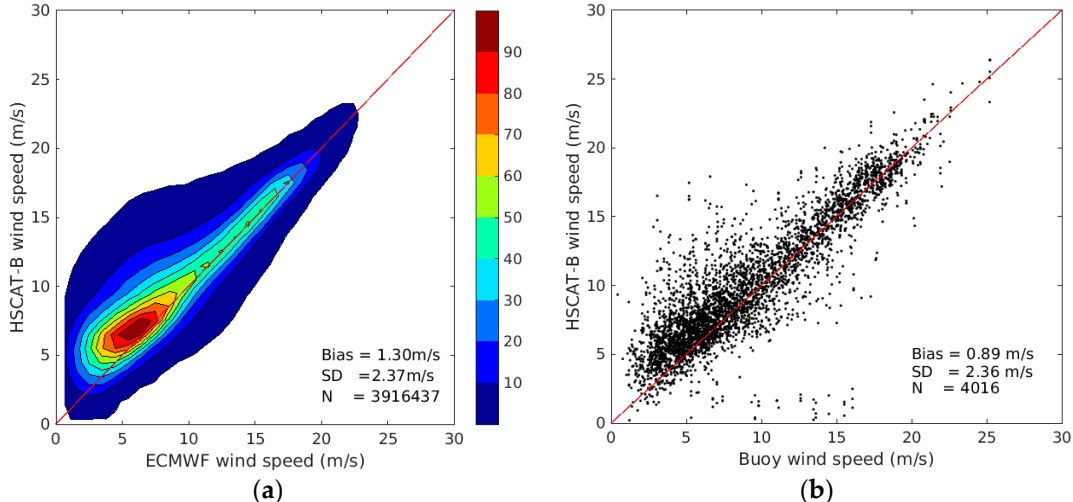

**Figure 2.** (**a**) Two-dimensional histogram of HSCAT-B QC-rejected data versus ECMWF winds (bins of 1 m/s), and (**b**) scatter plot of HSCAT-B QC-rejected data versus buoy wind speed. In (**a**), the color bar indicates the ratio (%) of data with respect to the maximum data bin, i.e., the lowest contour line level is 1%, and the following steps are 10%, 20%,..., and 90%.

### 3.2. Sensitivity of HSCAT Variables to Rain

The effects of rain on the HSCAT-derived winds were first assessed using the triply collocated HSCAT, ECMWF, and GMI data, due to the lack of triple collocations with HSCAT observations, buoy winds, and rain data. The main controversy of such assessment is the accuracy of ECMWF winds under rainy conditions. It was shown in [19] that the SD value of the difference between ECMWF and buoy wind speed increases remarkably with the rain rate, whereas the sensitivity of ECMWF wind speed biases to rain is much smaller than that of SD errors. On the other hand, the rain-induced wind speed biases of Ku-band scatterometers are much larger than the rain-induced ECMWF wind speed biases [20], hence the ECMWF background winds are used as reference when assessing the HSCAT wind speed bias under rainy conditions. Afterwards, the sensitivity of HSCAT QC indicators to rain was evaluated, with the objective of improving the rain flag, as well as the discrimination of valuable HSCAT observations.

Figure 3 shows the HSCAT-B wind speed bias w.r.t. ECMWF as a function of the averaged HSCAT-B and ECMWF wind speeds (a), and the CTD under different rainy conditions. The discrepancy between the HSCAT-B and ECMWF winds is remarkable for GMI rain rate (RR) > 1 mm/h and $w < {\sim}15$ m/s; in these cases, the volume scattering from rain cells may dominate the total radar backscatters, leading to positive HSCAT speed bias. For rain rates below 6 mm/h, HSCAT-B wind speeds above 17 m/s are generally in good agreement with the ECMWF reference, indicating that the rain attenuation and intensification effects on the scatterometer backscatters are more or less compensated. However, a large discrepancy is noticeable for RR > 6 mm/h, probably because that radar backscatter signal mostly comes from the rain layer rather than from the sea surface, such

that no wind information can be derived from the HSCAT observations. Figure 3b shows that the wind speed bias of HSCAT-B varies across the sub-satellite track, notably for the outer-swath and the inner-swath WVCs. This confirms that wind QC and quality assessment should be carried out for different WVC number (see Section 2.2).

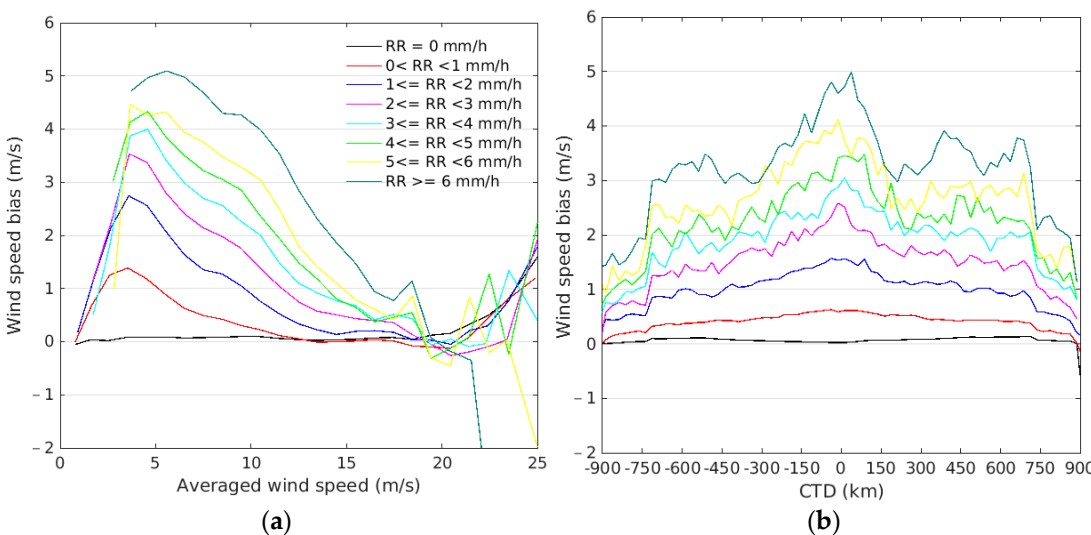

**Figure 3.** Bias of HSCAT-B wind speed (w.r.t. ECMWF winds) as a function of (**a**) the averaged HSCAT-B and ECMWF wind speed and (**b**) the cross-track distance to the sub-satellite track. Curves in different colors indicate different rain rates, as shown in the legend.

Following the results of Figure 3, the wind QC for HSCATs mainly intends to filter out the cases with RR > 1 mm/h (on GMI grid resolution). Moreover, the QC development is performed separately for the outer-, sweet-, and inner-swaths. Taking the sweet swath data as an example, Figure 4a,b show the percentile of rain-contaminated data (GMI RR above 1 mm/h) as a function of the HSCAT-B wind speed and the sorted percentiles by MLE and $MLE_m$, respectively. Figure 4c illustrates the same plot as Figure 4b but for the RapidScat [6], with purpose of comparing the sensitivity $MLE_m$ to rain between different sensors. The operational MLE-based QC threshold is converted into a rejection ratio and plotted in white-dashed curves. As such, the wind retrievals with speed and MLE on the left side of the dashed curve (Figure 4a) are actually rejected by the QC scheme.

The main conclusions and discussions are summarized as follows:

(1)　Compared to Figure 4a, a larger accumulation of rainy data appears on left side of the white-dashed curve in Figure 4b, so $MLE_m$ is even more effective than MLE in terms of flagging rain.

(2)　The current QC threshold is not optimal, as it rejects too many high winds without rain contamination (see the dark blue area on the left side of the dashed curve).

(3)　The QC indicator of HSCAT-D, either MLE or $MLE_m$, is much less sensitive to rain than that of HSCAT-B/C (not shown). As such, more data are rejected by the HSCAT-D QC scheme in order to ensure the quality of its QC-accepted winds (see Tables 2 and 3).

(4)　The $MLE_m$ of RapidScat is more sensitive to rain than that of HSCAT, particularly for high wind conditions. In other words, large $MLE_m$ values of HSCAT-B are less likely to be induced by rain compared to those of RapidScat. The main reasons for this could be: first, that HSCATs observe the Earth's surface at lower incidence angles (VV-48.5° and HH-41.5°) than RapidScat (VV-56° and HH-49°), such that their backscatters are less strongly affected by rain [19]; second, that HSCAT wind retrieval uses radar footprints, whereas RapidScat uses high resolution measurements, namely, slices [21].

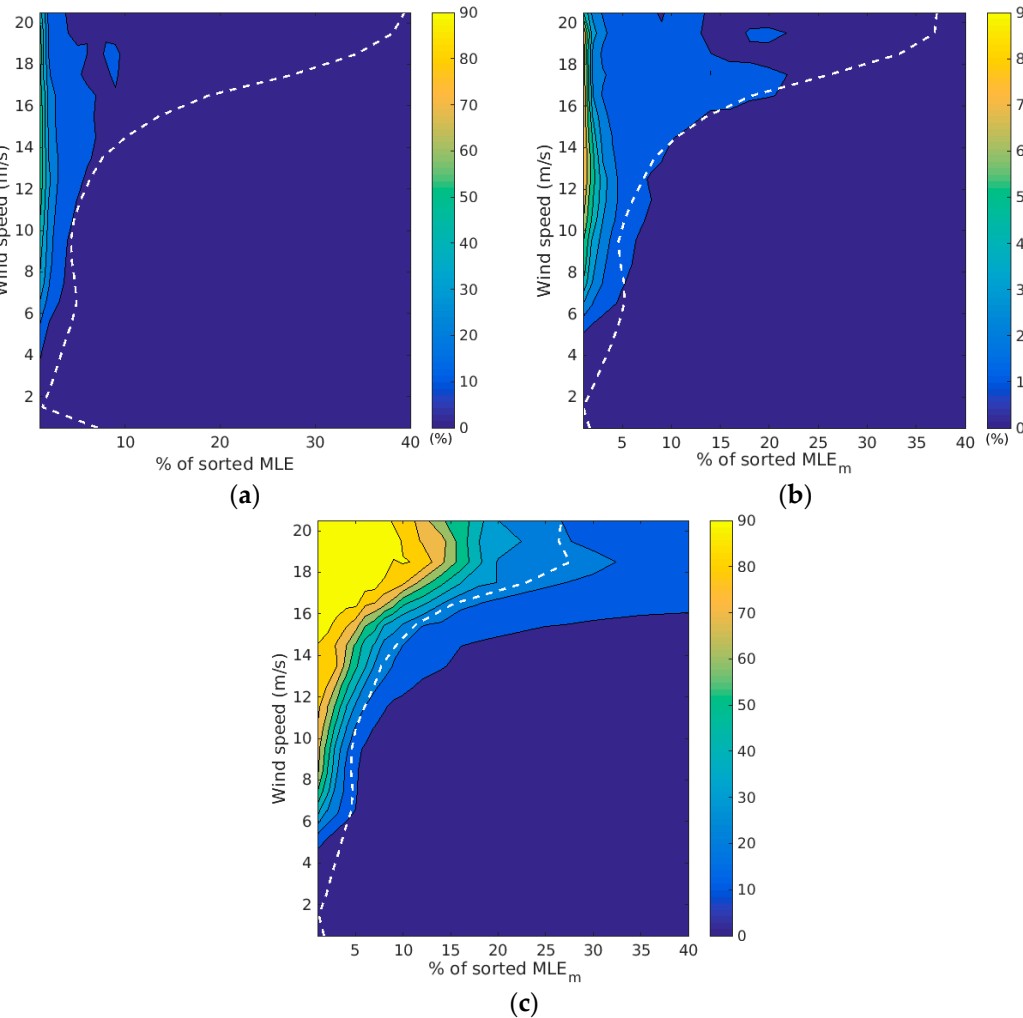

**Figure 4.** Percentage of rain-contaminated data (GMI RR > 1 mm/h) as a function of HSCAT-B wind speed and the sorted percentiles by (**a**) MLE and (**b**) MLE$_m$. (**c**) Shows the same plot as (**b**), but for RapidScat (figure adapted from [6]).The white dashed curve indicates the rejection ratio of the operational MLE-based QC.

## 4. Improved QC with MLE$_m$

### 4.1. Development of the MLE$_m$ QC Threshold

In this section, the MLE$_m$ is used to improve the HSCATs wind QC, notably for high winds. Similarly to Figure 4c, the key aspect of QC development is to determine of a wind-speed-dependent MLE$_m$ threshold, which aligns well with the isoline of the rain-contamination ratio, and, in turn, effectively separates the rainy observations from the "rain-free" ones. In practice, a new high-wind rejection rate for the percentiles corresponding to a rain contamination rate (RR > 1 mm/h) of 15~20% is proposed for the outer, sweet, and inner swaths, as indicated by the red-dashed curves in Figure 5.

A constant rejection rate is for wind speeds above 20 m/s, assuming that the MLE$_m$ sensitivity to rain is constant for $w > 20$ m/s. Note that the rejection rate for winds below 10 m/s is adapted slightly to keep the red-dashed curves smooth. In summary, the proposed method leads to a rejection rate of 1% for winds below 4 m/s, and of 5–10% (depending on the swath region) for winds at 20 m/s and above. More perspectives are summarized as follows:

(1) Similarly to the previous pencil-beam scatterometers, it is challenging to obtain a wind QC for the outer-swath WVCs, due to the lack of observations by the HH beam.

As such, on the left side of the dashed curves, the false alarm rate of rainy WVCs is higher than those of the sweet and inner swaths.

(2) The QC threshold is actually chosen by compromising the false alarm of rain in the QC rejected category and the missing alarm of rain in the QC accepted data. The wind retrieval bias induced by rain not only depends on the sea-surface wind condition, but also depends on the radar incidence angles [19]. Consequently, the rejection rate should be carefully adjusted for different sensors at various wind speed conditions.

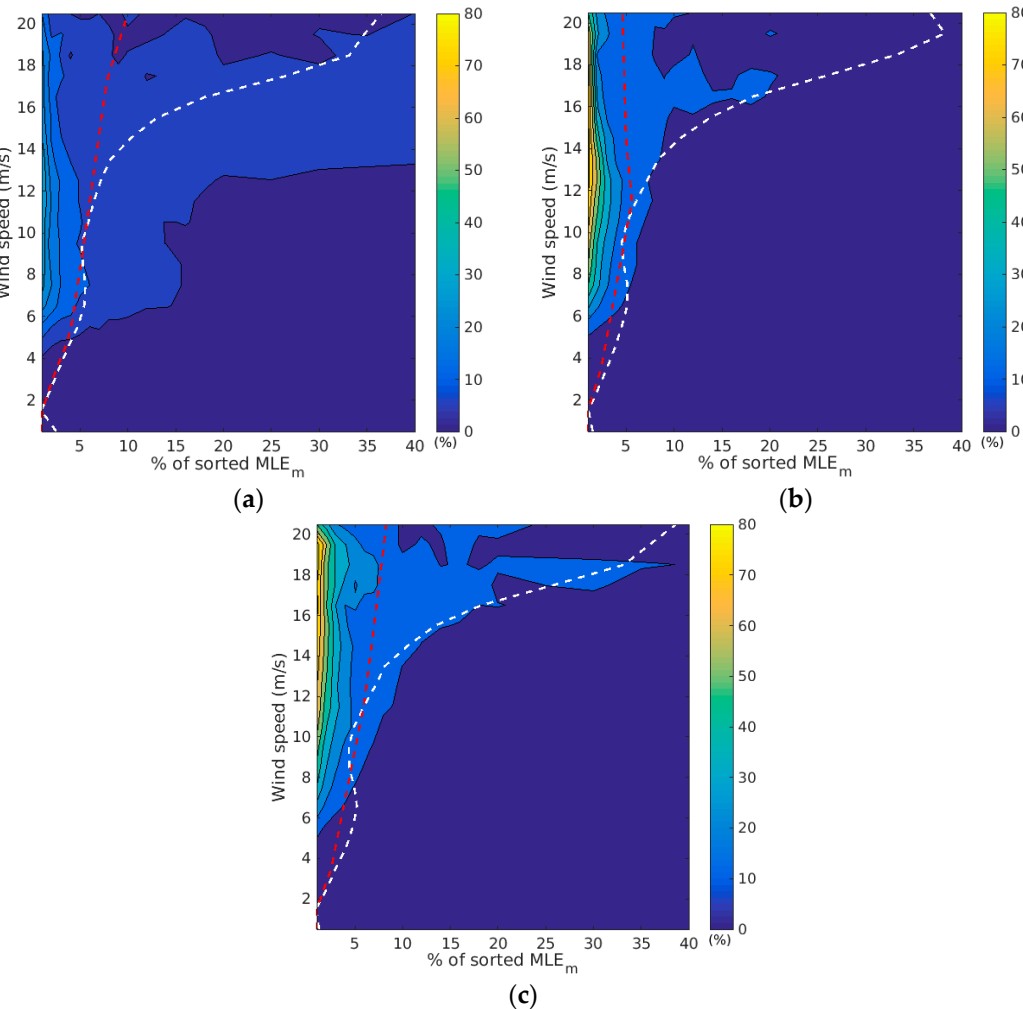

**Figure 5.** Percentage of rain-contaminated data (GMI RR > 1 mm/h) as a function HSCAT-B wind speed and the sorted percentiles by $MLE_m$: (**a**) outer swath; (**b**) sweet swath; (**c**) inner swath. Red-dashed curves depict the proposed QC rejection ratio as a function of wind speed, whereas white-dashed curves indicate the operational rejection ratio.

### 4.2. Validation of New QC

Afterwards, the performance of the $MLE_m$-based QC was assessed using different datasets, i.e., three months (October–December 2021) of the HSCAT-B L2B data, and three years of HSCAT-B and buoy collocations. Since the wind data users are more interested in the difference between the operational QC and the MLEm-based method, the following section mainly evaluates the wind data rejected by the $MLE_m$-based QC but preserved by the operational one (denoted as "QC1"), as well as those rejected by the MLE-based QC but preserved by the proposed $MLE_m$-based QC (denoted as "QC2"). Figure 6a,b show the 2-D histograms of HSCAT-B winds versus ECMWF background winds for the categories QC1 and QC2, respectively. As expected, the proposed $MLE_m$-based QC filters out 1.67% of 'poor-quality' data that are not detected by the operational QC, and saves a mass of high

winds which are consistent with the ECMWF reference. Figure 6c further validates the QC performance using the collocated HSCAT-B and buoy data. Compared to the MLE-based QC, the new QC demonstrates a remarkable mitigation of the over-rejection rates at high wind conditions.

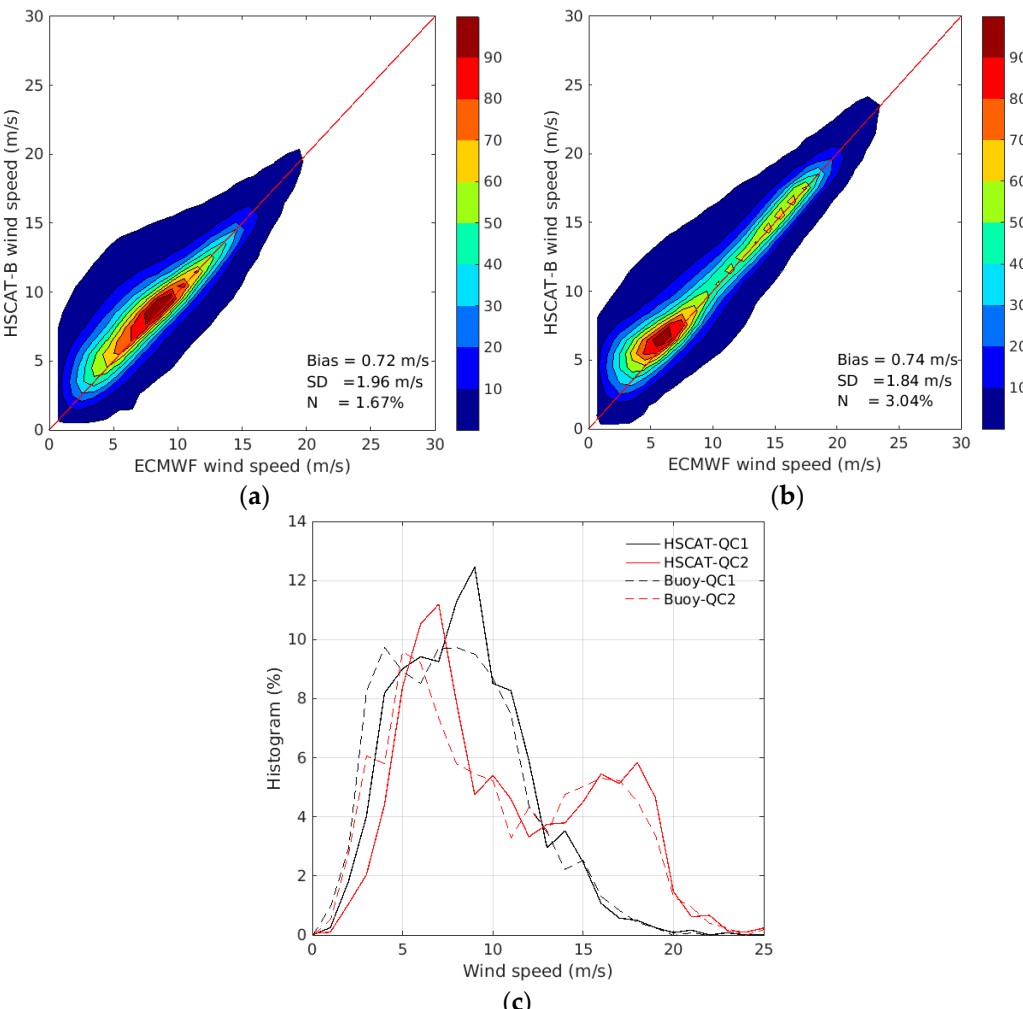

**Figure 6.** 2-D histograms of HSCAT-B winds rejected by the $MLE_m$ QC but preserved by the MLE QC (i.e., QC1) (**a**), and those rejected by the MLE QC but preserved by the $MLE_m$ QC (i.e., QC2) (**b**). The 1-D histograms of wind speed for categories "QC1" and "QC2" are shown in (**c**). In (**a**,**b**), the color bars have the same definitions as in Figure 2a.

However, Figure 6b indicates that the speed bias is still remarkable for winds below 8 m/s in the category QC2, probably due to the unfiltered rain-contaminated data. Since the objective of this paper is to improve the high wind QC for HSCATs simply based on $MLE_m$, we do not intend to further adapt the QC threshold for low and moderate wind conditions. Considering that MLE and $MLE_m$ are complementary for the QC's purposes, it is proposed that QC is performed for the entire wind speed region following these steps:

(1)    Reject the data whose $MLE_m$ values are higher than the QC threshold, i.e., on the left side of the red-dashed curves (Figure 5);

(2)    Reject the data with $w < 8$ m/s in the category "QC2".

In this way, the overall rejection ratio of the new QC is similar to the operational MLE-based QC. Finally, the above methodology was adapted for HSCAT-C and -D, and the statistical scores of the new QC are re-evaluated in Tables 4 and 5. Despite the fact that the statistical scores in Tables 4 and 5 do not show significant improvement, the high

wind rejection ratio, e.g., $w > 20$ m/s, is reduced from ~35% to ~8% (HSCAT-B and -C) or ~12% (HSCAT-D). Specifically, the performance of high winds' QC for the three HSCATs is summarized in Table 6. Due to the lack of HSCAT–buoy collocations, only the QC-accepted data are evaluated in this table. Generally, the proposed QC method keeps more and higher quality high wind data than the MLE-based QC.

**Table 4.** The same as Table 2 (w.r.t. ECMWF), but using the above new QC.

| Statistical Scores | | Speed (m/s) | Direction (°) | $u$ (m/s) | $v$ (m/s) | Rejection Ratio (%) |
|---|---|---|---|---|---|---|
| QC accepted data | HSCAT-B | 1.13 (0.09) | 10.6 (0.5) | 1.25 (0.08) | 1.22 (0.02) | - |
| | HSCAT-C | 1.07 (0.11) | 10.8 (0.7) | 1.26 (0.11) | 1.25 (0.01) | |
| | HSCAT-D | 1.09 (0.17) | 10.3 (0.4) | 1.22 (0.13) | 1.23 (0.03) | |
| QC rejected data | HSCAT-B | 2.40 (1.30) | 18.8 (−0.1) | 2.54 (0.42) | 2.28 (0.05) | 5.61 |
| | HSCAT-C | 2.26 (1.17) | 18.6 (−0.1) | 2.42 (0.40) | 2.24 (0.03) | 5.40 |
| | HSCAT-D | 1.93 (0.77) | 14.3 (0.0) | 1.94 (0.27) | 1.82 (0.04) | 11.6 |

**Table 5.** The same as Table 3 (w.r.t. buoy winds), but using the above new QC.

| Statistical Scores | | Speed (m/s) | Direction (°) | $u$ (m/s) | $v$ (m/s) | Rejection Ratio (%) |
|---|---|---|---|---|---|---|
| QC accepted data | HSCAT-B | 0.93 (0.14) | 13.6 (1.4) | 1.47 (−0.07) | 1.39 (−0.10) | - |
| | HSCAT-C | 1.11 (0.28) | 14.8 (−0.1) | 1.68 (−0.08) | 1.68 (−0.08) | |
| | HSCAT-D | 1.03 (0.17) | 13.8 (−1.0) | 1.49 (−0.01) | 1.56 (−0.07) | |
| QC rejected data | HSCAT-B | 2.34 (0.86) | 32.6 (2.2) | 3.47 (0.27) | 3.09 (−0.27) | 7.47 |
| | HSCAT-C | 2.14 (0.60) | 27.0 (−1.9) | 3.10 (−0.13) | 3.04 (−0.54) | 6.38 |
| | HSCAT-D | 1.94 (0.65) | 24.6 (−1.1) | 2.70 (−0.38) | 3.19 (−0.32) | 11.80 |

**Table 6.** The quality of QC-accepted of high winds ($w > 20$ m/s) w.r.t. buoy winds.

| Statistical Scores | | Speed (m/s) | Direction (°) | $u$ (m/s) | $v$ (m/s) | Number |
|---|---|---|---|---|---|---|
| Operational QC | HSCAT-B | 1.53 (−0.39) | 20.8 (−0.6) | 5.73 (−2.60) | 5.77 (−0.13) | 60 |
| | HSCAT-C | 1.45 (−0.41) | 12.1 (−0.8) | 3.46 (−0.54) | 3.16 (−1.20) | 64 |
| | HSCAT-D | 1.44 (−1.29) | 9.6 (−0.1) | 1.86 (0.62) | 3.23 (−2.02) | 26 |
| New MLE$_m$-based QC | HSCAT-B | 1.51 (−0.39) | 20.5 (−0.3) | 5.93 (−2.65) | 5.01 (−0.30) | 95 |
| | HSCAT-C | 1.57 (−0.32) | 16.1 (−0.9) | 4.98 (−1.10) | 3.47 (−0.66) | 103 |
| | HSCAT-D | 1.62 (−0.73) | 9.3 (0.3) | 1.72 (0.42) | 3.37 (−1.61) | 40 |

For the sake of comparison, the statistical scores for RapidScat are presented in Tables 5 and 6 as well. It shows that the HSCATs (except for HSCAT-D) generally reject the similar amount of data as RapidScat. Moreover, the wind quality of HSCATs is slightly better than that of RapidScat, as compared to buoy measurements. Under high wind conditions (e.g., $w > 20$ m/s), the MLE$_m$-based QC of RapidScat does not reduce the rejection ratio remarkably (>30%), which is in line with Figure 4c. Nonetheless, it is demonstrated that MLE$_m$ is better than MLE in terms of wind quality control for both HSCATs and RapidScat.

Finally, Figure 7 illustrates the results of two different QC indicators through a tropical cyclone observed by HSCAT-B on 24 October 2021, at 02:59. For illustrative purpose, only one arrow is shown within a 2 × 2 WVC box. Again, this proves that the new QC is effective for the HSCAT QC at high wind conditions: the operational QC rejects most of the high winds surrounding the cyclone center, which are of great significance to nowcasting, disaster warnings, numerical weather predictions, etc., whereas the new QC retains most wind vectors of the cyclone's structure.

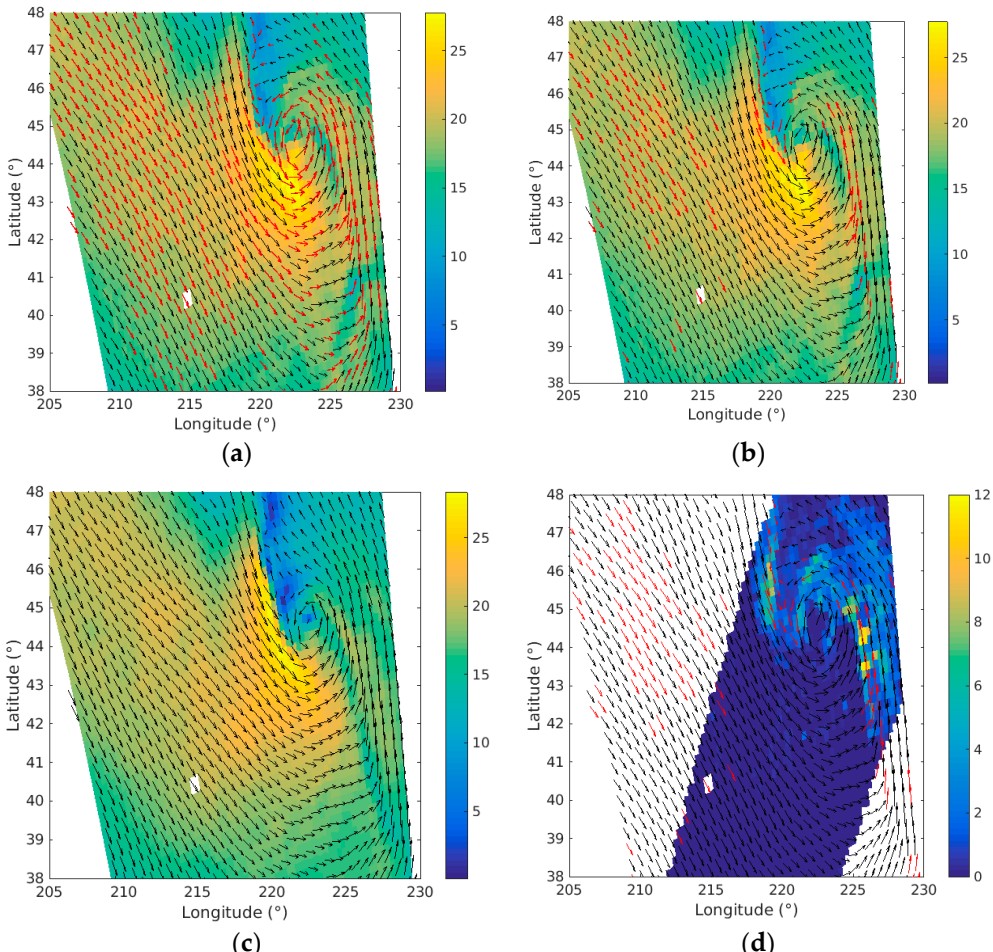

**Figure 7.** HSCAT-B wind vectors superimposed on the map of wind speed. Red arrows indicate the winds rejected by the operational MLE-based QC (**a**) and the proposed new QC (**b**). The ECMWF background wind field is shown in (**c**), and the collocated GMI RR is shown in (**d**).

## 5. Conclusions

The operational MLE-based QC of HSCATs, which was inherited from the QuikSCAT mission, generally discerns well between good- and poor-quality HSCAT winds. As such, sea surface winds acquired by the HSCATs are normally accurate, and consistent between the three sensors onboard the HY2-B, -C and -D satellites, providing unprecedented and valuable observations for those studying ocean winds. However, an overview of the HSCAT wind QC shows that more than 35% of high wind retrievals ($w > 20$ m/s) are rejected by the current QC approach, resulting in a great loss of valuable wind information, e.g., wind vectors surrounding cyclone centers.

Since rainfall is the primary factor in degrading the wind quality of Ku-band scatterometers, we investigated the sensitivity of MLE, as well as its spatially-averaged value $MLE_m$, to rain. It was found that $MLE_m$ is more effective than MLE in terms of flagging rain. Moreover, the HSCAT QC indicators are less sensitive to rain compared to the RapiScat on the International Space Station, probably due to the fact that HSCATs are operated at lower incidence angles (VV-48.5° and HH-41.5°) than RapidScat (VV-56° and HH-49°), meaning that their backscatters are less affected by rain, according to [19]. On the other hand, the HSCATs' wind retrieval uses radar footprints, whereas RapidScat uses high resolution slice measurements. As such, rain effects on the HSCATs' wind retrieval are rather well compensated under high wind conditions, providing an opportunity to reduce the rejection ratio of the high wind observations. Finally, a $MLE_m$-based QC is proposed to improve the

high wind QC for the HSCATs, which saves 23–28% of high wind retrievals ($w > 20$ m/s) that are originally rejected by the operational QC.

Compared to recent advances in Ku-band wind QC [11], the overall rejection ratio of the new QC is still slightly too high, especially for winds below 10 m/s. Consequently, future work will focus on exploiting other quality-sensitive parameters (e.g., SE and Joss) for HSCATs QC purposes, and to verify whether the combination of different QC indicators is able to better discrimination rain-induced poor-quality winds from the good-quality observations. Moreover, the sensitivity of HSCAT-D QC indicators to rain needs to be further investigated, in order to achieve more consistent wind quality among the three HSCATs.

**Author Contributions:** Conceptualization, W.L.; methodology, S.L., Y.Z. and Y.J.; software and validation, W.L. and S.L.; formal analysis, W.L.; writing—review and editing, S.L. and W.L. All authors have read and agreed to the published version of the manuscript.

**Funding:** This research was funded by the National Natural Science Foundation of China under Grant 42027805 and Grant 42192561.

**Data Availability Statement:** Not applicable.

**Acknowledgments:** The authors would like to acknowledge NSOAS for providing straightforward and rapid access to the HY-2B data. We would also like to acknowledge Remote Sensing Systems for providing the GMI data.

**Conflicts of Interest:** The authors declare no conflict of interest.

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
