# Peer review of "On the Quality Control of HY-2 Scatterometer High Winds"

_remotesensing, doi:10.3390/rs14215565_

Round 1
Reviewer 1 Report
The manuscript try to investigate the effect of the spatially averaged MLE (MLEm) on the quality control (QC) of HY-2 scatterometer under high wind conditions. The MLEm has been successfully applied to improve the RapidScat wind QC. The results did show some differences in high winds QC for the two methods. However, there are still some unclear issues which need to be clarified before it can be accepted for publication in RS.
Comments
1. The main purpose of the manuscript seems to be the QC improvement under high wind conditions. However, all the comparisons between the new and old QC method mainly focus on the normal wind speed. The results under high winds should be proposed for all the figures and tables. I understand that there are not many data for high winds. But the improvement for normal wind conditions are not significant for the new QC method. Therefore, the authors should focus on the differences under high wind conditions.
2. For the wind speed over 20m/s, the ECMWF winds may not be a good reference. The CFS (CFsV2 or CFSR) Winds may has less bias under high wind conditions.
3. The spatial distribution of buoy has a greater impact on the credibility of the evaluation. Please add the distribution of buoy to Figure 1 or the appropriate Figure..
4. Please show the colorbar for Fig.2 and Fig.6.
5. Lines 207-209: please rewrite this sentence, it is confusing.
6. Fig.7: There are no significant differences in (a) and (b), please add wind distributions before QC and the ECMWF. The case in Fig.7 shows few high winds (> 20 m/s) around the TC. It should be changed into a much stronger case.
7. Which dataset were used in the comparison, ERA5 or ERA-I?
8. Line 177, the font styles are different for u and v.
9. “HSCATD” should be “HSCAT-D” in all tables.
10. “Rejection ratio” is confusing. Is it the percentage of rejected data?
11. Lines 184-185: refereces should be added here.
12. rain rate(RR) and cross-track distance (CTD) show twice in the manuscript.
13. Table 4: the rejection ratio significantly increases for HSCAT-D. Why?
Author Response
Dear reviewer, thank you very much for your vaulable comments. Please find our response in the attached file.

Reviewer 2 Report
The main idea of this manuscript is to improve HSCATs winds QC of HY-2 satellite series that flagging rain data by substituting spatially averaged maximum likelihood estimator (MLEm) for maximum likelihood estimator (MLE). There are already several prior studies of Ku-band scatterometers that applied this method. This study is to verify whether it works on HY-2 or not.
The main works of this study include: comparing (i) the standard deviations of the difference between HSCATs and ECMWF winds and (ii) the standard deviations of the difference between HSCATs and the mean wind vectors of collocated buoy data, with both the condition under primary QC and new QC.
This manuscript has following contributions: First, although the results do not show significant improvement of the statistical score, but the rejection rate of high wind (>20m/s) is reduced from ~35% to ~8-12%, and point out 1.67% of “poor quality” data that are not alarmed by the operational QC, conserving a great mass of valuable wind information such as cyclone from original rejection of operational QC.
Only minor revision is needed before accepted for publication.
1. A comprehensive review or explanations are needed of why HSCAT-D has obviously different statistic scores and rejection ratio to that of HSCAT-B&C.
2. Reference performance statistics of MLEm QC results such as RapidScat, QuikSCAT are necessary in extra table for comparison.
Author Response

(The authors gave the same response as above.)

Reviewer 3 Report
Comments to “On the quality control of HY-2 scatterometer high winds”
This study evaluated the performances of quality control schemes based on the maximum likelihood estimator (MLE) and its spatially averaged value (MLEm) to HY-2 scatterometer high winds. Compared to the ECMWF and buoy winds, it is found that MLE is too conservative and rejects up to ~35% of high winds. Therefore, MLEm was set to improve the the high wind quality control. MLEm is more effective than MLE in terms of flagging rain data and mitigates the over-rejection at high wind conditions.
This interesting study has proposed a new quality control method for scatterometer winds. It can be used to provide higher-quality satellite winds, which has great significance for nowcasting, disaster monitoring and numerical weather prediction data assimilation. In general, the manuscript is well organized and written. However, I have some concerns listed below and a mandatory revision is needed.
Main comments:
1 Ln 169-170: ECMWF and buoy winds were used as references to evaluate the performances of quality control schemes on HY-2 winds. Have you ever compared the results of HY-2 winds with other satellite winds (e.g. WindSat and CCMP) ?
2 The rejection ratio of HSCAT-D is much higher than HSCAT-B and HSCAT-C. However, only the result of HSCAT-B were shown in this study. Please add some analysis on HSCAT-D to make the evaluation more convincing.
3 Figures 4a and 4b showed that MLEm mitigates the over-rejection at high wind conditions. However, at low and moderate wind conditions, a few rain-contaminated data was accepted in MLEm (see the light blue area on the right side of the dashed curve). Is it indicated that MLEm has poorer performance than MLE at low and moderate wind conditions?
4 Tables 4 and 5: Why is the rejection ratio of HSCAT-D in MLEm even higher than that in MLE?
5 Have you compared the performances of MLEm in different oceans?
6 Is this new quality control schemes (MLEm ) available for other satellite?
Specific comments:
Fig 1b: Please adjust the range of colorbar to make the result more clear.
Table 2, 3, 4 and 5: “HSCATD” => “HSCAT-D”
Ln 218: “RR” => “GMI rain rate (RR)”
Ln 245: “GMI rain rate (RR)”- Move the full name of RR to its first appearance at Ln 218.
Author Response

(The authors gave the same response as above.)

Round 2
Reviewer 3 Report
I appreciate the author's detailed replies to my comments. Now I am satisfied with the current version.
Author Response
Thank you again for your kind review on the manuscript.